# Stakeholder Challenges and Opportunities of GPS Shock Collars to Achieve Optimum Welfare in a Conservation or Farm Setting

**DOI:** 10.3390/ani13193084

**Published:** 2023-10-03

**Authors:** Iris Alexandra McCormick, Jessica Elizabeth Stokes

**Affiliations:** School of Agriculture, Food and the Environment, Royal Agricultural University, Cirencester GL7 6JS, UK; jessica.stokes@rau.ac.uk

**Keywords:** animal welfare, cattle, electric collars, grazing, conservation, virtual livestock fence/ing

## Abstract

**Simple Summary:**

This study aims to understand how virtual livestock fences (VFs), introduced to the UK in 2020, were being used and to identify the animal welfare opportunities and challenges of VF systems. VF systems work by training livestock to respect a virtual fence line by pairing an audible warning tone with an electric shock. Semistructured interviews were conducted with key stakeholders (n = 8), including policymakers, leading conservation managers, and regenerative farmers. We found VF systems were utilised in three main settings: 1. Extensive grassland, where physical fences were prohibited or uneconomical, such as national parks or mountains; 2. Conservation grazing, where livestock grazing is primarily used to manage habitat for biodiversity, such as nature reserves; 3. In mob, strip, or rotational grazing systems, where VFs replace physical electric fencing to reduce workload. This study finds that, in extensive settings, VFs can help safeguard livestock from dangers and aid stockpersons to locate and care for animals. However, in restricted settings, VFs are less predictable and controllable for livestock than physical fences, resulting in livestock (cattle) receiving a greater number of electric shocks than experienced with physical electric fences. The use of VFs in a restricted setting poses a significant risk to livestock welfare. Protocols are needed to encourage best practices and help prevent the misuse of VF systems.

**Abstract:**

Virtual fences for livestock facilitated by a GPS shock collar (GPS-SC) and phone app were introduced to the UK in cattle herd trials in 2020. Technology which uses aversive shocks to control livestock movement on farms and in other settings poses a significant risk to livestock welfare. There are currently no welfare protocols in place in the UK to ensure the ethical use of GPS-SCs. The objective of this study was to understand how GPS-SCs were being used in practice in the UK and gather data to assist researchers and policymakers in the future research and development of a welfare protocol for the UK. We studied how the technology performs in terms of welfare challenges and opportunities, covering extensive livestock production, conservation settings, “rewilding”, and regenerative farming practices, where the technology is currently being applied. Semistructured interviews were conducted with key stakeholders. In-depth interviews (n = 8) supported the previous literature that the use of GPS-SCs in restricted grazing settings poses a risk to animal welfare. This is due to the wavering virtual fence boundary line (which is affected by satellite movements), a lack of visual markers, and, in some “rewilding” and conservation settings, livestock keepers, which require training and support to enable optimal welfare in practice and prevent misuse of the technology. Results also indicated that there are opportunities for enhancing livestock welfare with GPS-SCs in very extensive farm settings, where targeted care can be facilitated by using the data to monitor and track livestock using GPS-SCs, and which can also prevent cattle injury or fatality through virtual pastures designed to protect livestock from hazards such as roads or bogs. Future research is needed to focus on minimising shocks in the training period and to better understand the value of visual electric fences in the training process.

## 1. Introduction 

GPS shock collars (GPS-SCs) for cattle have been gradually introduced to the UK market over the last 3 years through the Nofence© company (Batnfjordsøra, Norway) in trials on farms and in conservation grazing settings to address three main opportunities for fencing grazing livestock [1,2]. Firstly, to address the challenges of fencing unenclosed land and to extend grazing into unfenced or difficult-to-fence extensive grazing land, such as the uplands of Scotland and northern England [1]. Secondly, to manage cattle in rotational, strip, or “mob” grazing systems, where the infrastructure and labour required for the regular movement of fences is costly and time-consuming. Thirdly, for conservation grazing systems, where livestock are introduced to restore or improve biodiversity across a range of habitats, including seminatural calcareous grassland, woodland, forests, and some riparian habitats [1].

European countries are developing policies to move towards rewarding farmers for ecosystem services, such as the European Green Deal and the Environmental Land Management (ELM) Scheme, developed in England [3,4,5]. Grazing with livestock can assist in the restoration and conservation of grassland, heathland, wood pasture, as well as in coastal and floodplain settings by grazing marsh, fen, scrub and scrublands, and saltmarsh and sand dunes [6,7]. Light grazing and autumn grazing benefit many species, including rare butterflies [8] and insect and plant diversity [5,9]. The collars are being trialled and proving successful for the management of these areas opening across different parts of the UK [2,10]. Controlled grazing regimes are increasingly being advocated for livestock grazing systems to improve soil health and sward recovery times [11]. Visual electric fences are mostly used, but farmers are beginning to consider virtual fences (VFs) with GPS-SCs [2]. This enables the pasture to be modified quickly using the app rather than the manual task of moving electrical fencing.

As a baseline, the welfare opportunities and challenges of VF systems can be assessed using the Five Freedoms [12]. GPS-SCs, such as Shepherd^TM^ (Australia), and Nofence©, have the potential to facilitate mob and other intensive grazing systems [2], or may be used to herd cattle [13], which can fail to meet the five freedoms. Both of these practices have the potential to evoke fear, distress, and pain where cows are repeatedly shocked (Freedom 3 and Freedom 5), and restrict normal behaviour (Freedom 4). However, the potential welfare benefits of using VF systems to facilitate additional extensive grazing can offer livestock more grazing choices and the freedom to express normal behaviour (Freedom 4). The GPS technology can also be seen to help livestock keepers locate cattle more easily to reduce the chances of untreated disease or injury (Freedom 3), and promotes good pasture design to prevent injury (or death) on roads or around hazards such as bogs (Freedom 3). Welfare-outcome-assessment protocols for beef cattle could assess VF systems [14] and data from GPS-SCs could provide automated verifiable evidence of welfare outcomes [2]. Nofence© provides users with these data as part of their service and alerts users to livestock escapes. 

Most experiments to assess both the efficacy of GPS-SCs and the welfare of livestock using this technology to measure and evaluate stress experienced by livestock exposed to electric shock use the cognitive theory of stress (CATS) [15,16]. Therefore, if livestock fitted with GPS-SCs can easily predict and control their interactions with the VF, this will help minimise the negative welfare associated with receiving aversive shocks [15,16,17,18]. In contrast, predictable but uncontrollable experiences cause pain, fear, and distress (Freedoms 3 and 5). Therefore, where animals fitted with GPS-SCs are unable to avoid electric shocks through poor virtual-pasture design [2,19], using the system to drive livestock [13] in tightly packed virtual pastures where mob grazing is practised, or due to faulty equipment and wavering GPS signals, the welfare consequences can be pain, fear, and distress (Freedoms 3 and 5). However, evidence for social learning in cows can facilitate the avoidance of shocks in both visual electric and virtually fenced systems [19,20]. In an historic experiment with visual electric fences, approximately 50% of animals across the different groups tested did not receive a shock during the training phase [20]. In VF fields, cattle also showed social-learning behaviour [19,21,22]. However, these results varied significantly across groups. Despite social learning taking place, cattle do not appear reliably and predictably able to avoid shocks in VF fields, and continue to receive shocks despite taking cues from the behaviour of other animals [7]. 

Some cows were also observed to confidently lead whilst others were timider and reacted quickly by “running away” when hearing other cows’ audio warnings [7]. The ratio of audio cues to shocks also varies according to herd dynamics, with some individuals rarely receiving both audio warnings or shocks, whilst other animals appear bolder, regularly receiving both audio warnings and shocks [7]. More research is needed to assess the impact of personality on the welfare outcomes of VFs [23]. 

When cattle are contained with visual electric fencing, research demonstrates socially learned avoidance [20], suggesting that some cattle can learn to respect the fence without receiving a shock. This experiment also demonstrated that cattle were easily able to generalise their learning to recognise and respect physical electric fences in other locations. By comparison, in VF systems trials, all cattle trained to avoid the VF with audio cues experienced a shock at least once [18,22]. In an experiment to compare visual electric tape with the VF, 28.49% of the encounters with audio warnings resulted in a shock over the 4-week trial [18]. In an experiment with mob-grazed dairy cows, 39.8% of audio warnings resulted in a shock, where cows received on average 1–6.5 shocks per day [22,24]. Furthermore, results from the trial to compare the visual electric fence and VF took place in simply designed, rectangular paddocks with one virtual line [18], and not in a more complex, real-life, paddock design, where cattle were contained on all boundaries by the VF, and sometimes with exclusion zones included in the design [2,7,25]. These results indicate a significant welfare insult, which violates freedom from pain, fear, and distress (Freedoms 3 and 5), especially in mob, strip, rotational grazing, and conservation grazing systems.

Within the groups tested, the number of shocks experienced before learning to associate and respond correctly by stopping or turning was on average 2.5 shocks, and ranged between 1 and 6 shocks per audio warning [18] and 1–6.5 shocks per audio warning [22]. The eShepherd^TM^ website information also confirms that each animal will need to receive between five and seven shocks to learn to associate the audio cue with the electric shock. The greater number of shocks experienced by livestock in VF systems compared to visual electric fences is a welfare concern [24,25].

Both McSweeney et al. (2020) [25] and Umstatter (2015) [26] observe that cattle recognise and understand virtual fence systems (VFSs) more easily if linked with a visual cue. With VF GPS-SCs used together with visual cues, cows quickly understood the boundary [26]. Dairy cows in the McSweeney (2020) [25] experiment showed signs of stress, measured by reduced grazing and ruminating behaviour, in VF blocks of grazing without visual cues. 

The Electronic Collar Manufacturers Association™ (ECMA, Orleans, France) have developed voluntary codes of conduct in the use of shock collars to contain dogs. They explicitly require visual markers to help dogs recognise a boundary by sight, as well as through an auditory warning. Future research should therefore focus on the use of visual fences to optimise welfare as part of VF training protocols for livestock. There is no recent study directly comparing the two fencing systems [7]. Therefore, more research is needed to better compare visual and virtual electric fences to reduce or eliminate shocks in VF systems so to improve cattle welfare where they are applied in a conservation or farm setting.

The Five Freedoms predominantly focus on reducing negative welfare: 1. Freedom from hunger and thirst; 2. Freedom from discomfort; 3. Freedom from pain, injury, or disease; 4. Freedom to express normal behaviour; 5. Freedom from fear and distress [12]. Recognising animals have positive welfare experiences is now enshrined in policy through the Animal Welfare (Sentience) Act 2022 [27] and a strategic objective of industry [28] to advance quality of life or a “good life” for farm animals [29,30,31]. The “Good Life” concept has been applied to develop a framework for assessing positive welfare [32], which could also be utilised in VF systems to evaluate the potential opportunities, challenges, and welfare trade-offs of using GPS-SCs. Pasture-based systems provide positive welfare opportunities of play, breeding and nurturing, food enrichment, pasture choices, positive social interactions within the herd, an enriched environment, and a natural body type, or telos [32]. However, challenges include challenging conditions (such as mountainous, unimproved grazing lands, and poor weather conditions) where inadequate forage and suboptimal thermal comfort may be experienced. Furthermore, facilitating mob, strip, and rotational grazing systems with VF systems can come into conflict with shade and shelter opportunities for livestock, or their preferences for resting places [31,32,33].

The authors cannot find evidence of research, guidelines, or policy relating to the number or strength of electric shocks considered “acceptable” on welfare grounds during training or post-training. In contrast, the Electric Collar Manufacturers Association (ECMA) have formulated a voluntary code of conduct for dogs and cats using containment collars [34]. Shock collars used for dogs and cats have been outlawed in Wales since 2010 under the Animal Welfare Regulations [35]. Eight other European countries, as well as parts of Australia, have also banned collars on welfare grounds; however, they are still allowed in England, even though there are calls for a ban in Scotland [36,37]. 

The Nofence© VF system emits an aversive shock at 0.2 Joules at 3 kV for 1.0 s [2]. Researchers testing similar GPS-SCs in Australia using eShepherd^TM^ found that 800 volts in pulses of less than 1 s was successful for containing cattle [16,21]. Current studies use faecal cortisol metabolite assessments and behaviour patterns, such as speed back to grazing, to measure stress in livestock [18,38,39].

High shock levels can lead to adverse reactions in cattle, expressed as lunging forward and bawling [40]. Most studies with cattle do not report adverse reactions to shock; however, there is evidence that some individuals (goats and cows) do not learn to turn when they receive the audio cue and shock, but instead become immobile [41,42]. Minimising shock levels whilst maintaining an aversive deterrent is the goal [2,18,39]. Lines et al. (2013) have developed a stimulus-strength-ranking indicator (SSRI) to assess the impact in a range of shock collars for dogs that could be adapted and applied in a livestock setting [43].

There has been a clear call for the regulation of GPS-SCs to ensure the welfare of livestock, standards for use, and the training of stock people together with independent monitoring to ensure welfare compliance [44]. The authors advocate that a clear set of verifiable protocols for the use of this technology are needed to ensure the government’s priorities for improving the welfare of cattle at pasture [45]. Furthermore, there is currently no independent training for stock people, farmers, and land managers using this technology [3,44,45], nor monitoring systems to assess the welfare implications in practice [2,3].

As the environmental land-management scheme (ELMS) is rolled out alongside the Animal Health and Welfare Pathway [45,46], farmers are incentivised to deliver both environmental and animal welfare enhancements as public goods. Therefore, technology that can facilitate environmental goods cannot be to the detriment of animal welfare goods [47]. This technology is still in the development process, and there is no regulatory framework or training standard in the UK, whereas the roll-out on farms is becoming more popular [3].

This study therefore sought to: 1. Understand how users are currently applying the technology in practice; 2. Discover the key stakeholder opportunities to enhance sustainable husbandry systems; 3. Understand the key welfare challenges of applying GPS-SC technology in extensive, regenerative, and conservation grazing settings that need to be overcome to improve cattle welfare at pasture [45]. To be able to scale up this technology ethically, it is important that welfare training protocols are codesigned with researchers and practitioners to not only optimise the value of environmental land management without a detriment to animal welfare, but to move beyond safeguarding animal welfare outcomes and toward optimising positive welfare opportunities in line with government and industry policy.

## 2. Methodology

### 2.1. Recruitment

Existing farmers, land managers, and other stakeholders with practical experience were targeted to evaluate how the GPS-SC works in the field in relation to livestock welfare. As the technology is currently expensive and in development, the number of candidates did not equate to wider data collection using quantitative methods. Possible candidates were defined as those with experience in using the collars in practice, who tended to be leaders or innovators in their field, including a range of stakeholders from farmers and conservation land managers to policymakers or developers of the technology. In addition, welfare stakeholders are defined as those with expertise in livestock management or leaders in their field who have in-depth knowledge of livestock welfare. As such, qualitative methods to understand a range of stakeholders’ attitudes, behaviours, and experiences were utilised. Recruitment was conducted using both networks known to the authors via email or telephone and research by the first author. As the purpose of this study was to identify a range of attitudes, experiences, and behaviours associated with GPS-SC use, rather than the relative frequencies of these across a representative sample of stakeholders, this sampling strategy was deemed appropriate. Ten possible participants were contacted by phone and email to request an interview, with 80% of potential candidates interviewed (n = 8). A profile of interviewees can be found in Table 1.

### 2.2. Interview Process

Semistructured interviews were carried out to ensure standardisation across stakeholders, as well as the flexibility to follow up and explore the themes that arose. Open questions were chosen to allow interviewees to talk in-depth, but with a guided focus to enable the interviewer some opportunity to compare and evaluate feedback based on a set of iterative questions [46,47]. A practice interview took place to assess the length of the interview and number of questions that would fit into a practical timeframe. During this practice interview, the recording and transcribing app was tested on both the phone call and online video calls. There was some flexibility around which questions were delivered to different stakeholders [46]. However, all relevant questions were asked in four sections to aid consistent data collection and analysis (Table 2). Interviewees were sent participant information sheets and a consent form prior to the interview. All interview dates and times were arranged in advance and took place in March 2022. Interviews were conducted online via Teams (4), by phone (n = 1), or in person (n = 3), and were recorded via Otter.ai. Interviews lasted 20–30 min. The interviewer listened carefully to the answers and remained encouraging but did not lead the participant nor enter into dialogue during data collection.

### 2.3. Data Analysis

After the semistructured interview had taken place, the recordings were transcribed and read in detail to ensure the clarity of the transcription. All data were then entered into NVivo for thematic analysis. Themes were then identified by the first author through open coding, which searched for patterns as well as variance in stakeholders’ answers with regards to attitudes, experiences, and behaviours associated with use of GPSSCs, and the welfare challenges and opportunities associated with the technology. Analysis of the transcripts found 11 emergent themes. To assist with clarity in presenting the results, participants were numbered and coded as follows: Livestock Manager (LM); Policymaker (P); Technology Developer company (D). For example, Participant 5 is a livestock manager; therefore, their code is P5.LM.

## 3. Results and Discussion

### 3.1. Livestock Welfare Expertise

The first theme to emerge in relation to animal welfare and the application of GPS-SCs on-farm was participants’ experience and expertise with animals. Two participants did not have any farming (P6.LM) or animal expertise (P3.D), but had become extremely interested in the GPS-SC technology because they valued the environmental benefits and wanted to work with the company who was developing the technology:

*My engagement comes from a soil health and climate perspective* (P3.D). 

As one participant (P6.LM) stated, they saw the cows:

*As a tool* (P6.LM).

This was in the context of conservation work and restoring biodiversity within a project working to re-establish the seminatural calcareous grassland on common land. A further indication that GPS-SCs are being applied in practice and increasing in popularity in a conservation setting was confirmed by P7.P, who said:

*It’s (GPS-SC’s) becoming more and more of a thing—people buying up big estates and doing rewilding* (P7.P).

This suggests that some people applying this technology may be new entrants, novel livestock keepers, and GPS-SC users. It is therefore important that training in livestock husbandry, behaviour, and welfare is delivered in conjunction with GPS-SC training. For example, P6.LM said they had:

*Recently done a day training course on animal welfare for stock checking* (P6.L).

Previously, P6.LM had not taken much interest in the welfare of the cattle, saying of the conservation group:

*We’re not stock people* (P6.LM).

However, since the course, this stakeholder expressed an interest in animal behaviour and welfare, where they had learnt the five freedoms and how to observe livestock. For example:

*(We learnt to) approach them from a distance and watch them before they know you are there to see them interacting* (P6.LM).

This theme indicates that the provision of protocols for GPS-SC technology should be included together with husbandry, behaviour, and welfare training, and would benefit users and facilitate optimum livestock welfare, especially in conservation grazing settings where users are potentially less likely to have livestock education. Verifiable welfare assessments, protocols, and stock-keeper training are key government policy goals [27] for improving cattle welfare at pasture [45]. Furthermore, extensive grazing can provide positive welfare opportunities to cattle [32], and future training protocols should include not only the five freedoms as a baseline, but positive welfare, which is now the focus of both government strategy, through the Animal Sentience Bill [27], and industry strategy, through the Agriculture, Horticulture Development Board (AHDB) and Pasture for Life [28,48,49].

### 3.2. Welfare Benefits in Extensive Settings

The second theme that emerged from the data focused on the welfare benefits when using GPS-SCs in extensive grazing and conservation settings. There was a consensus that this technology was, with some exceptions, beneficial for livestock management, and welfare benefits included tracking livestock and being able to exclude livestock from hazardous areas, which supported the previous literature [10] that GPS-SCs can be advantageous in extensive grazing systems. The following findings will therefore be discussed in the context of the five freedoms and the trade-offs between (environmental land) management and optimal welfare, defined by the “Good Life” framework.

In extensive systems, there was less mention by participants of the shocks associated with the collars, and a clear emphasis on the welfare benefits. These included being able to track livestock in remote areas and target care. For example, stakeholder P7.P could see the benefits:

*A big problem in actually checking stock daily and keeping within standard requirements—but often they can’t find them—so absolutely, if they had GPS collars, then you know, you can track them down in that sort of extensive environment and check them as you need to* (P7.P).

The same stakeholder also noted that rewilding and extensive systems do:

*Have problems with quite high mortality, because stock isn’t checked as regularly* (P7.P).

Another stakeholder also highlighted the convenience of GPS-SCs for managing animals in conservation grazing settings:

*GPS tracking from a time management point of view, would make you check them far more easily. Therefore, you would do it more regularly, whereas I think in some extensive situations, they’re not monitored as much as they might be. And particularly in sort of conservation grazing situations where you’ve potentially got, you know, ditches and valleys and wet areas and you know, if animals get stuck or any of those sorts of problems, then clearly being able to easily find them. It’s very much of a of a benefit* (P7.P).

Furthermore, all LMs commented on how much the app provided them with novel information about livestock movements and time budgets, which offered insights into cattle behaviour. For instance, looking at the data on the app, P5.LM said:

*I think I have learnt a bit more about animal behaviour from watching them move with the collars on* (P5.LM).

This stakeholder went on to demonstrate on the app how they can identify the changes in activity before, during, and after the birth of a calf. Understanding animal behaviour, and being able to track it on the app, could not only help prevent suffering and pick up on diseased or sick animals early [12], but also monitor behaviour to enhance positive welfare opportunities [32]. Using GPS-SC data to improve the welfare of cattle at pasture that support natural behaviours associated with grazing and being outdoors can also support in the delivery of Defra’s (2023) animal health and welfare priorities [5].

Most participants also explicitly talked about how very large pastures, by definition, ensured the cattle rarely encountered the VF, and this was considered a welfare benefit by LMs. As an illustration of this, P5.LM said:

*The bigger the paddock the better and the only time we get issues is when they are in a small area, when you are trying to train them—as soon as they are in a big open space, and they can express their behaviour more naturally the thing (GPS-SC) works better* (P5.LM).

There was a clear consensus that it was normal and natural for cattle to be grazing and outside, and that GPS-SC technology enables an extension of grazing and browsing into hitherto unfenced or unfenceable areas, which potentially provides animals with novelty and choice. A *more varied diet* was cited by participants as a welfare benefit of using the technology to extend grazing. This was seen as a positive outcome for both nature recovery and livestock health and welfare. Research, however, has not been conducted to establish whether cattle in these settings are indeed provided with the added opportunities of novel food they would choose and prefer [32], and, more fundamentally, whether these settings always provide the basic nutritional requirements for cattle.

Three participants identified that the animals could express other natural behaviours beyond grazing, such as scratching, choosing a more comfortable place to rest and cud, and seeking out different shelter or shade, tantamount to the positive welfare opportunities [32]. For example, one stakeholder expressed:

*You can see them enjoying the sun on the south bank and then go into the trees or hollows to get out of the weather and they love to find a tree to scratch* (P6.LM).

Another stakeholder also identified several other positive welfare opportunities as a result of observing the cattle in this environment:

*I’ve learned that they like to rub a lot on things and they like a variety of diet* (P4.LM).

These observations by users of GPS-SCs support several of the “Good Life” objectives for cattle—a comfortable physical and thermal environment, as well as an enriched environment which provides novelty and choice [32]. There are also data supporting the “Good Life” objectives of the Food Enrichment and Pasture choices [32]. For example, observing the different plants conservation grazing cattle prefer where GPS-SCs facilitate wood-pasture creation, one stakeholder said:

*They love the old man’s beard and they seem to go for the ivy* (P5.LM).

Likewise, another stakeholder (P6.LM), talking about a moving pasture using a hay net to lead the cattle, observed that *ivy* served just as well as a tasty inducement.

### 3.3. Welfare Issues in Extensive Settings

However, there were also trade-offs in terms of welfare in extensive and seminatural settings, highlighted by one stakeholder, due to the natural environment and consequent abundance of flies and midges. P5.LM noted that, in conservation settings, the cattle suffered more from flies and biting insects than on conventional farmland, saying:

*I treat them more often with fly repellent than at my father’s farm* (P5.LM).

This could lead to unnecessary suffering and discomfort [12], and goes against the “Good Life” objective of comfortable physical environment [32]. Interestingly, other participants (n = 3) also had concerns about the potential loss of daily handling and observation by farmers, stock-keepers, and land managers that is facilitated by the technology:

*So, I think that’s really important (positive interactions with the cattle through daily handling) and it is a temptation to, for example, you can move the animals from your bed, and allocate them to a new pasture, but I think there’s no replacement for Ground Truthing that in the flesh on the ground* (P4.LM).

This observation suggests a trade-off with the “Good Life” objective’s positive experience with stock-keepers [32], and that stock-keepers perceive the observation of cattle as very much part of their role and purpose. Furthermore, regular handling is practical for livestock management. Cattle that are tame and relaxed around the handler and handling equipment, such as trailers and cattle crushes, have been shown to have reduced pain and distress during routine injections and examinations [50].

Finally, a lack of animal experience and expertise, which has an impact on understanding animal behaviour and welfare, was noted by participants (n = 2) as a potential concern in “*rewilding*” settings. They observed that rewilding projects were increasing and that sometimes livestock are introduced into these environments and treated more like wild animals with little or no husbandry intervention. Stakeholders quoted anecdotal evidence that this was justified under the banner of being “*natural*” and as part of the process of rewilding. The welfare concerns were expressed as follows:

*I think it’s only going to become more of a thing, in terms of you know, this rewilding environment, but having responsibility for the animals that you then put in that environment is, you know, this really is really key, the idea you just sort of let them go off and look after themselves. I’m afraid it’s not quite good enough* (P7P).

Overall, stakeholders suggest that extensive livestock systems would benefit from the GPS-SCs. This is in line with the previous literature [3,10]. However, the data also suggest that animal husbandry, behaviour, and welfare training protocols would be useful to ensure the new technology is used ethically, and livestock keepers have the right kind of support to ensure the success of their rewilding projects and the enhancement of livestock welfare [27,45].

### 3.4. Welfare Issues in Intensive Grazing Settings

The fourth theme which emerged was welfare issues in intensive grazing settings. Most (n = 5) of the participants highlighted how the collars could be used in a farming system with mob, strip, or rotational grazing schemes. Participants (n = 2) commented how it would save time and labour moving the pasture with the aid of the phone app. However, half of the participants (n = 4) had reservations regarding livestock welfare in these settings due to the lack of visual markers as an added stress for cattle already herded together and compelled to graze closer to the boundary to obtain fresh grass. There was a sense from participants (n = 3) that, in this setting, the GPS-SCs would be more likely to cause welfare issues. For example, one stakeholder said:

*For mob grazing uses, the collar is, maybe a bit too much kind of a pressure in terms of welfare—I think it’s probably really annoying to the animals where in intensive operations with daily or twice daily moves, the areas are quite small and visually because there wasn’t a visual on it. They can see an electric fence but with collar boundaries, they can’t see it and they’re constantly being harassed by this beeping noise, and if they ignore it they get an electric pulse* (P4.LM).

Another stakeholder (P5.LM) also had concerns about the technology being used in this setting, as the cows *push and shove* each other in close proximity. They were also concerned that there will be welfare challenges associated with an unacceptable number of shocks and stress in an otherwise already pressured environment for the cattle. This is supported by previous reports [3,44].

All of the LMs observed that the boundary, which is not fixed, as with a visual fence, but moves around due to the satellite technology, is problematic from a welfare point of view:

*The boundary moves, moves and wavers, doesn’t it? It’s not a sort of set line* (P5.LM).

One LM also noted that this was disconcerting for the cattle:

*But the GPS signal does just wander around a little bit so you can get a situation where they are stood grazing perfectly still and happy, and then they haven’t moved but they get a warning. And they just have to turn around because the signal has sort of moved towards them* (P6.LM).

It was also noted by some participants that cattle prefer not to interact with the fence. However, one LM noted that, when the forage is getting scarce in a paddock, the animals begin to interact more with the boundary:

*If they can they avoid going near the boundary* (P5.LM).

A wavering line together with confined spaces could impact animal welfare, causing pain, fear, and distress [12], and does not provide a comfortable physical environment [32]. A further observation by one LM highlighted that a positive interaction with stock-keepers happened when the cattle were moved between pastures manually. Comparing the daily moving of visual electric fences with the VF, they said:

*When I’m operating behind an electric fence I’m signalling to those animals, I’m calling them they’re getting a cue. They can see what I’m doing. So, they know, they associate that action, those cues with fresh grazing and a sort of positive reward and something as being safe. It’s difficult to recreate that in a virtual sense* (P4.LM).

This experience supports the existing literature [50] that creating positive associations with rewards in an otherwise stressful or painful situation can lessen the fear and pain experienced by livestock. With a VF, there is no need to be near the livestock to change the pasture if the user knows the terrain and fencing plan well enough; therefore, the chances for positive interactions with the stock-keeper can reduce with the use of the technology.

To summarise, one LM talked about the lack of visual cue, the wavering boundary line, and the necessity for cattle to interact with the boundary to obtain the fresh grass, saying:

*I think that does totally mess with the psyche of the animal* (P4.LM).

These experiences, highlighted by participants, confirm that there are greater risks to livestock welfare in a confined grazing setting where GPS-SCs are used to facilitate mob, strip, rotational, and some conservation grazing settings. A clear set of protocols need to be developed to ensure there is a training regime and post-training protocol which minimises shocks, so to ensure welfare is guaranteed and in line with government policy [27].

### 3.5. The Value of Training

There was a wide consensus around the importance of successful, comprehensible training for cattle or other livestock where GPS-SCs are used. As highlighted in the existing literature, there was broad agreement between the participants that the process had to ensure that cattle find the system predictable and controllable to ensure good welfare [16]. For example, one participant said:

*The virtual fence has to be predictable and controllable for the animal in order for us to say that, you know, we’re happy with the animal welfare* (P3.D).

In support of this assumption, another participant stated:

*We need to be sure that animals understand what the audio means and that they can avoid the discomfort of the pulse. And we need to do everything we can to help that with our system* (P2.D).

These statements show that technology companies understand the importance of comprehensive training and work towards ensuring predictability and controllability for the livestock by monitoring the outcomes of collar use remotely as part of their service. For example, if an animal escapes from the virtual paddock, the user is sent a text message. And crucially, if there are *too many* shocks or escapes, the user is followed-up with a call from the company. All LMs reported that interactions with the app and text message alerting them of escapes worked well. However, there is no existing research on the impact of shocks from a welfare perspective and, consequently, what is, for example, an acceptable threshold. Despite existing literature reporting that some animals are not able to learn how to interact with the VF [41,42], none of the participants in this study reported untrainable cattle. For example, one stakeholder stated:

*If there’s anything wrong, then we would look at the ratio between audios and pulses. And if that wasn’t very equal, then it’s evident that the animal isn’t understanding what it’s supposed to do, but we haven’t had any animal never understand* (P3.D).

This finding may be due to improvements in the design of the technology or training advice given by the company, or due to the small sample size of participants in the current study. The latter was not properly understood [20], and further research in this area is needed. In the next theme, the value of visual cues is discussed in more detail.

### 3.6. Visual Cues

Visual cues were mentioned mostly by the land manager participants interviewed (n = 4). Some participants deliberated if there should be visual cues to aid cattle comprehension of the VF system and thereby reduce the number of interactions and shocks the cattle received. For example, one LM said:

*I think the other thing really is about those visual cues. I’ve always tried to set my virtual boundaries somewhere near resembling physical features. So, a hedgerow or a fence or something that the animals can associate the signals they’re getting, with what they can see. Because I think otherwise, they’re happily grazing along in the field and they get the beeping wondering, what on earth did I just move into?—It’s a bit like a bird flying into a glass window* (P4.LM).

Similarly, another participant stated:

*There needs to be visual lines for them to be able to associate the audio alarm with the boundary* (P7.P).

This knowledge and experience from practitioners support the existing literature [25,26,51] that visual cues aid and optimise cattle learning of VFs. Therefore, further research should investigate this, and policymakers should work with practitioners to develop welfare protocols where visual cues can optimise cattle training. This may also apply to settings where space is restricted or where boundary *desire lines* (P5LM) are used by livestock.

### 3.7. Advancing Training for Optimal Welfare

This theme draws on the experience and insight of one participant (P5.LM), who suggested a modification to the way cattle are trained to reduce the number of shocks experienced during the initial contact with the VF using both visual cues and a change to the design of the training paddock. Based on an experiment this practitioner did with cattle untrained to the collars, they found that the herd can be contained in a field with gaps in the fence without the cattle being shocked (Figure 1):

*So some I haven’t trained the cows and used the collars just to secure real fences where there is a weak spot and they have learnt the system without going through—running through—but when you put the line right across an open field they don’t necessarily learn anything from getting 3 zaps* (P5.LM).

The participant explained that this test showed that, on this occasion, the cows turned away when they encountered the VF without running through, even though they had no prior training, suggesting that a visual cue aided their learning process. In relation to reducing the number of shocks, the participant suggests:

*There is something to be said, from a welfare point of view potentially, if they can just gently interact with the boundary* (P5.LM).

This practitioner suggested setting the VF boundary along the visual fence, but 5 m in from the visual fence rather than across the open part of the field, according to the instruction manual. In describing the system they plan to use this year (2022), they stated:

*Put the VF just 5 metres inside the boundary and then they won’t be able to run through it and get 3 shocks. I think they can learn just as well from getting one shock and then trying it again rather than running through it ... and getting beep zap, beep zap, beep zap escape, I don’t think they learn much from that it is just the way the manual is written to train them* (P5.LM).

The interview with this practitioner supports the conditioning theory and associative learning theory [16], and could reduce the welfare challenges of GPS-SCs associated with training cattle to understand the VF. There is evidence that other VF systems use visual fixed fences in the training period [51]. These insights and experiences from practitioners in the field demonstrate the value of cocreating optimal training protocols for welfare enhancement in collaboration with existing farmers utilising the technology. These training suggestions should be included in training protocols with VF systems.

“Belts”, “zaps”, “pulses”, or “shocks”?

This theme explored the language used by different stakeholders and revealed a difference in the perception of pain experienced by the animals. It was found that the technology developers consistently use the word “*pulse*” on their website, and this behaviour was supported in the interviews. For example, one D using the word “*pulse*” talked about the experience as “*uncomfortable*” and preferred not to use electric “*shock*”, as it suggested a more painful experience.

On the other hand, one LM familiar with visual electric fencing consistently used the more prevalent and, arguably, stronger farm term “*zap*”. This participant also raised the issue of audio warnings and zaps directly at the start of the interview, consistently using the word “*zap*”, and was very keen to show exactly how many “*zaps*” each animal had had. For example:

*Look, this one has had 63 to 1 and this one has had 36 to 2* (P5.LM).

By contrast, another LM using the word “*pulse*” was less concerned about the number of shocks a cow received, but made an observation about the combination of cues:

*The beeping noise and pulse was probably really annoying to the animals* (P4.LM).

This LM was more concerned about the overall experience of the GPS-SC, from when cows were fitted with the collars through the whole process in practice. In contrast, another LM was actively motivated to find ways to reduce the number of shocks cattle experienced during the training period through modifications to the VF line in training paddocks, as discussed in the theme advancing training for optimal welfare.

Using the word “pulse” can make the electric shock seem less painful than if the word “shock” or “zap” is used, and it was noted that, when one user reported accidently shocking themselves whilst handling the collars, they changed their vocabulary and used the word “belt” instead, describing the experience as:

*I got a hell of a belt off it!* (P8.LM)

Use of language has been shown to affect the way people perceive emotions and feelings in others [52]. As animals are nonverbal, we can only understand their feelings by observing their behaviour, emotions, and physiological reactions to stimuli [53]. This research has not been systematically performed in a VF setting. Furthermore, some of the current language used when applying GPC-SCs to cattle and other livestock can be seen to understate the pain and stress that an animal can feel when being shocked, which can lead to an acceptance of practices that cause animal suffering and, in the case of GPS-SCs, an indifference to the number of shocks cattle receive.

### 3.8. Personality

All the LMs showed a keen interest in how the data from GPS-SCs highlight animal behaviour and personality traits. As the data record all the movements of the herd and individuals within it, this provides valuable information about how cows interact with each other and the VF, as well as other resources in the environment. Social learning, as evidenced in previous literature [18,19,22], was confirmed by all participants (n = 8). For example, one participant stated:


*Some animals just will get shocked more than others. Some will learn from others and never get shocked. Like*


*100 and 112 they won’t go anywhere—just hear the others, follow the others, and just hear the beep and turn around. Whereas others just have so many more interactions with the boundary* (P5.LM).

Personality differences were also highlighted by the participants, supporting the existing literature [7,23] where certain individuals within the herd received many more shocks than others. One participant stated:

*There are noticeably always the leaders and the followers* (P4.LM).

This variance has been identified as a welfare concern [23], and demonstrates a welfare challenge in terms of unnecessary discomfort or pain, thereby not meeting the five freedoms [12]. It was also highlighted by all the LMs that there are always one or two cows who push the boundary and get more shocks.

*Most of them are 34:1 but look here 63:3 that is Peanut, Peanut’s a troublemaker anyway* (P5.LM).

*One particular animal who is always out the front exploring—the kind of questions that go through my mind about how the herd dynamic is working and what’s going on in animal mind—who steps over the line and who doesn’t?* (P4.LM)

The findings from this theme support the previous literature, which suggests that optimum welfare could be maximised by choosing cattle with less adventurous personality traits when using VFs [23]. Protocols to manage this practical challenge would need to be developed in collaboration with practitioners.

### 3.9. Comparing Fences

Nearly all participants (n = 7) recognised that interactions between livestock and visual electric fences are not recorded, whereas every interaction with the VF is recorded and monitored, and this was a source of puzzlement to participants when asked to consider welfare and GPS-SCs. As a typical illustration, one LM observed after considering how it was difficult to compare the differences between visual electric fences and VFs:

*Nobody keeps any data on how many electric shocks cows get on a daily basis from a normal electric fencing unit* (P5.LM).

This observation supports the previous literature that there is limited research [20] and no recent comparative studies which look directly at how often cattle are shocked by visual electric fences. Studies comparing visual and virtual electric fences do not take this direct comparison into consideration when designing their experiments [16,17,18]. Replicating the McDonald (1981) [20] experiment with modern monitoring equipment would help inform our understanding of livestock learning and allow stakeholders to make informed comparisons about electric fence use and animal welfare. This is even more important in confined grazing, such as mob or strip grazing, where welfare is already a concern [25]. Shock collars used to stop dogs from straying are aided by visual flags or markers along the virtual boundary [34]. This should be investigated and, where valid, applied in a livestock setting.

Further research on the comparison between visual electric fences and VFs through monitoring cattle where visual electric fences are used would also inform users if, or to what extent, livestock learning can be accelerated by the introduction of visual aids. Accelerating the learning process and ensuring livestock receive the minimum number of shocks in the process is a welfare priority supported by all stakeholder participants in this study, and would enhance the welfare of cattle at pasture [27,45].

### 3.10. Towards a Welfare Protocol

There is currently no legislation around this technology, and this study observed the attitudes within the technology company reflecting this fact. For example, when asked if there was a minimum age for a calf to wear a collar, two Ds answered:

*We don’t have a set recommendation (P3.D)—we trust the farmer—he knows his animals best* (P2.D).

Despite the current lack of legislation, the technology company does provide plenty of advice and clear guidance [2], and this is supported by the data collected. For example, one D, responding to the question about the minimum age of cattle, added:

*We say (to the farmer) 1. Is the animal physically strong enough to handle the collar? and 2. Is the animal mentally mature enough to understand the system?* (P3.D)

Further evidence that the technology developer is committed to ensuring the best outcome was expressed by this statement:

*We have the back office system, it is where we catch up on any collar that isn’t performing the way we expect it to perform. And that allows us to sort of have that end-to-end control if there’s anything going wrong* (P3.D).

Typical problems encountered were expressed by one LM as follows:

*I make sure there is water and shade well within the boundary so they never have to interact with the boundary to get to water and shade* (P5.LM).

Hunger or thirst will drive cattle to escape the fence and interact with the boundary, incurring an unacceptable number of shocks. This was confirmed by one D, who said:

*The key bit is to remember enough feed, water and to be in an environment that the cattle are comfortable. So don’t sort of think that you can just put any cattle anywhere* (P2.D).

All stakeholders talked about designing the virtual paddocks in such a way as to preclude angles under 90 degrees, or narrow corridors where the cattle can become *stuck* between boundaries and encounter signals and shocks from all sides. All participants also reported that there was a short delay between switching off the collar to move the paddock, and problems would occur if the user forgot to switch off the collar before changing pastures or after loading up a trailer. These issues would lead to livestock being repeatedly shocked as they were driven in the trailer out of the VF boundary.

Animal welfare is a key government policy and industry aim [12,27,45]. Welfare assessment is essential to identify welfare issues [31] and reward good practice [54,55]. This section highlights some of the significant welfare challenges associated with GPS-SCs and confirms that protocols are needed to ensure they are used ethically to encourage optimal welfare. Welfare protocols drawn up in collaboration with the technology company, industry stakeholders, and welfare researchers, as identified in this study, are a minimal requirement. Furthermore, data from the app could be used to verify the ethical use of the technology in practice, as well as a monitoring scheme to measure and verify positive welfare outcomes.

### 3.11. Limitations of the Study

The recruitment method used within this study does not provide a statistically representative sample from each of the stakeholder groups. This means biases may have been introduced, particularly with regards to recruiting participants who were enthusiastic or had reservations about the technology, and therefore it is likely that similar and polarised experiences, attitudes, and perceptions are presented. In addition, demographic information of the interviewees, such as age, gender, and years of experience within their stakeholder group, were not reported, which limits our understanding of the study population and the context in which their views are positioned. Given the stated limitations of this study, these findings are not considered a comprehensive representation of the stakeholder challenges and opportunities of GPS-SCs to achieve optimum welfare in a conservation or farm setting. However, this study does highlight valuable, in practice, knowledge, which supports the findings from existing literature, and crucially highlights the areas for further research and policy development. The authors advocate for collaboration with farmers and land managers who are already implementing this technology to ensure the negative welfare consequences of GPS-SCs are minimised, and the positive welfare opportunities highlighted are realised in practice.

## 4. Conclusions

This study indicates that the provision of protocols for GPS-SC technology, together with other training and support, such as joint learning and assessment, would benefit users, facilitate optimum livestock welfare outcomes, and mitigate challenges to livestock welfare [53,54,55,56].

There is evidence to suggest that, in conservation grazing and rewilding settings, users are less likely to have livestock expertise. As the UK government is developing environmental land-management schemes (ELMSs) to reward land managers for ecosystem services, more people will be encouraged into this sector, where livestock are seen as key to achieving environmental objectives. Therefore, stock-keeper training and protocols for the use of GPS-SCs are a priority in these settings and support key government policy goals for animal welfare [27,45].

Stakeholders interviewed during this study state that, on balance, in very extensive grazing schemes, livestock welfare could be improved by the introduction of GPS-SCs [10]. This would enable targeted care for livestock through tracking on a mobile phone app whilst reducing the chances of livestock fatalities by designing safe virtual pastures on the app. Stakeholders also support the previous findings [25,26] that visual cues aid the cattle training of VFs. Future research and development must therefore focus on validating visual cues for VFs and, where valid, develop protocols which integrate visual cues as part of GPS-SC training, and for settings where space is restricted. Further research to understand how livestock interact with visual electric fences, to enable a better comparison with VFs, is required to assess to what extent livestock learning can be improved by the introduction of visual aids to optimise the training programme for GPS-SCs [20]. Findings from this study also support recent research highlighting welfare challenges associated with bold and explorative temperaments in cattle when applying GPS-SCs in practice. This could be minimised by choosing cattle with less adventurous personality traits for use with VFs [23]. However, the welfare impact on cattle with different temperaments in this setting requires further investigation.

Data from this study indicate how language describing electric shocks can influence the way people perceive animal pain or discomfort [52]. From a welfare and ethical perspective, it is important that language is used to accurately describe the welfare insult from the animal’s point of view. Further research to measure the welfare impact of electric shock collars to cattle, as conducted in other species, such as dogs, is pertinent to knowledge and education on the ground. Otherwise, we risk not understating the pain that animals feel, and, in the case of GPS-SCs, indifference to the number of shocks they receive. As part of this study, one practitioner has shared innovative ideas to reduce the number of shocks animals receive in the training period through trials on-farm. This highlights the value of engaging with practitioners in the field and working collaboratively with practitioners and industry to codesign policy which delivers both public goods for animal welfare and environmental land management [55].

## Figures and Tables

**Figure 1 animals-13-03084-f001:**
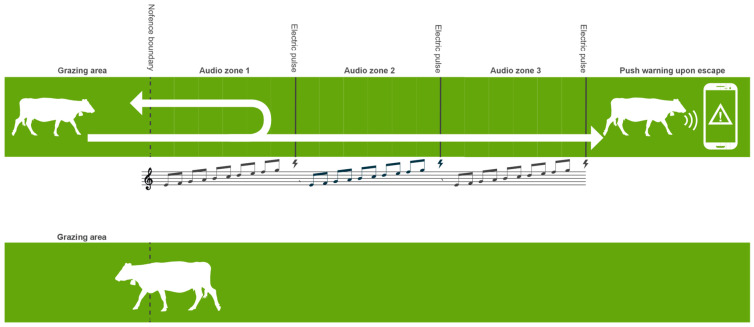
*Cow interacting with virtual fence*. Nofence©, 2023. Reprinted with permission.

**Table 1 animals-13-03084-t001:** Participant profiles.

Participant	Profile	Code
Farm Livestock Health and Welfare Consultant and Policy Advisor	Leading a team of health and welfare assessors, farm advisors, and policymakers for a farm assurance organisation	P7.P
Farm Manager	Managing livestock on an important regenerative farming project and rewilding centre. Formerly the general manager and founder of a farm assurance organisation	P4.LM
Farm Manager	Managing livestock for leading conservation charity and family farm.	P5.LM
Sales and Marketing Manager	GPS-SC developer leading the team to introduce GPS-SCs to the UK	P2.D
Sales and Marketing	GPS-SC developer	P3.D
Charity Trustee and Manager	Conservation-charity founder restoring calcareous grassland	P6.LM
Farmer	Sheep and cattle on an extensive farm	P1.LM
Charity Trustee and Farm Manager	Sheep and beef on an organic farm	P8.LM

**Table 2 animals-13-03084-t002:** Semistructured interview questions.

Interview Section	Questions
Introduction	How did you get into livestock farming? Tell me about how you became interested in virtual fencing?
Second section	What breed/age/of cattle are you using with the GPS-SCs? Where are you using the collars? How do you think the collars work best? Have you seen them working in practice and what did you think?
Third section	How do you define animal welfare? What are the opportunities and challenges to livestock welfare using collars?
Fourth section	Is there a place for codesigning policy with stakeholders and farmers using the collars?

## Data Availability

Raw data in the form of anonymised transcripts, as well as coded datasheets can be made available on request to the first author.

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
