# Peer review of "Stakeholder Challenges and Opportunities of GPS Shock Collars to Achieve Optimum Welfare in a Conservation or Farm Setting"

_animals, 2023, doi:10.3390/ani13193084_

Round 1

Reviewer 1 Report

This manuscript considers practical implementation of virtual fencing technology within the UK based primarily on Nofence devices (as far as I could tell). Considerations of practical management strategies and welfare implications are included. Semi-structured interviews were carried out with different stakeholders although across only a small number of participants. The results/discussion section is very long – some aspects are not completely novel, e.g., the section on personality/individual differences in interactions. This is displayed across a lot of the literature already so not sure if this could be summarised more succinctly within a different section.

The manuscript does come across as having somewhat of an animal welfare agenda negatively biased toward VF. For example, later on in the results/discussion section there is mention that we don’t actually know how many shocks cattle receive with standard electric fences. This is true, and to me is a critical point when highlighting all the negative impacts of the shocks with VF. How do we know it is more (or less?) than standard electric fences.

I’ve made more specific comments as below.

Line 9: the wording doesn’t quite make sense, seems like an incomplete sentence, are some words missing?

Simple summary: what about the literature review as mentioned in the abstract. If it is not a structured lit review and it is just a standard way of summarising the lit as part of an introduction then why state it is a lit review in the abstract?

Line 7 and throughout. Consistency in whether is it VF or VFs or VF’s. Similarly, double-check for consistency in GPS-SC vs GPSSC throughout

Line 15: aid stockpersons to locate

Line 34: and support to enable optimal

Line 47: so is this paper strictly about Nofence devices? If so, that should be made clear. As there are other VF products on the market and they all operate slightly differently in how they deliver the signals as well as I presume training information? Some of the training protocols ‘coming out of’ this paper, are already applied with other VF companies.

Line 48: I think it would read better as ‘Firstly, to address the challenges’

Line 69-70. Do you mean the available pasture area is quickly modified or are you trying to say the actual pasture is modified more rapidly because you can use the app to move animals?

Lines 74-75. Where is your evidence that these practices will ‘repeatedly shock’ the animals? Is the VF going to be substantially different from any other cattle management practice for these activities. E.g., herding with a vehicle, or dogs etc? How is freedom 4 (normal behaviour) specifically impacted by the use of VF that wouldn’t otherwise be impacted by other practices to the same means?

Line 82: could you provide more detail on automated data from GPS-SCs could be used to provide verifiable evidence of welfare outcomes?

Line 85 approx. I think it needs to be introduced much earlier on that the VF devices use audio cues to enable the animals to learn. There is only mention of the shocks until much later on which makes it read like these are just shock collars with no warning/learning component for the animals.

Line 90: is this supposed to be in reference to Lee et al 2018 that introduced this theory in relation to the CATS?

Lines 92-94: What is the eShepherd 2023 ref you are citing there? Does that specifically state that the system drives livestock? It is not designed to do that, that is not one of the intended uses of it. It prevents them from moving forward (and backwards if you place a fence behind them), but there is not a fence that drives animals forward from behind as you may with traditional herding techniques, I.e., it doesn’t shock them to make them move in a forward direction. Also where is the evidence about overcrowded VF pastures? Is this based on the literature since this section is supposed to be a literature review. Would this be different from overcrowding in a physically fenced paddock? Similarly with poor VF designs. What evidence in the literature do you have on this? Online user interfaces in some systems will specifically prevent the producer from implementing what would be considered a ‘poor design’, eg sharp corners.

Line 100: what is the Campbell et al. 2021 ref, I couldn’t see it in the ref list.

Line 103: what do you mean ‘continue to receive shocks’? I looked and this was only a 3-day study. This would encompass the learning period, so how can you conclude the animals didn’t reliably and predictably avoid shocks? It is assumed that for an animal to learn the audio association properly, they have to experience the shock themselves rather than just watching other animals after only hearing an audio tone of their own. That would be basic aversive learning principles that operate in the VF algorithm.

Line 107: see, this is the first proper mention of the audio warnings which I think is misleading to the reader that the VF collars are only shock collars.

Line 119: How do you know they were ‘not understood’ and ‘not controllable’. Did this trial incorporate the animals learning about the cues? In which case they are not going to know what the audio means until they have had a few associations of the audio with the shock. Did some animals choose to push into the boundary? Were there differences in pain sensitivity? How did you conclude that these animals didn’t understand. This is also where it is important to consider that we don’t know how many shocks animals receive with physical electric fencing. Maybe it is less, maybe the same, maybe more. This is a lot more challenging to measure given there are no records the way the collar devices record, which is possibly why studies to date doing comparisons have not included this metric.

Line 143: but other VF products do use the physical fence when they are first training the animals, this is part of their training protocol. Also what about all the literature showing successful VF learning without visual cues? The audio cue replaces the visual cue, that is the benefit of the system. What about where VF fences are moved and the cattle can still respond appropriately showing they are moving around based on the audio cue rather than a visual stimulus?

Line 179-180: Exactly, these adverse reactions are an indication that welfare would be compromised. But many studies have specifically stated they watch when the animals are first learning to ensure there are no adverse reactions, and that the shock level has been determined based on minimising adverse reactions (because animals won’t learn properly when their reactions are too strong). So doesn’t that mean there has been welfare consideration of the shock level? Maybe not number of shocks, but it is incorrect to say no one has considered what the shock level should be in terms of the animals’ welfare.

Also what about faecal cortisol metabolite assessments that have occurred across several VF studies, and assessments of behavioural patterns. This text implies that no one has considered welfare impacts of VF but that is simply untrue. This is what makes the introduction seemed biased and not an accurate representation of what has been published.

Line 206: again, this text reads like welfare has not yet been considered in designing VF systems and the training required. This is really not true at all.

Line 229: can be seen in Table 1.

Line 251: This sentence doesn’t seem grammatically correct, are there some words missing from it?

Line 252: All data were

Section 3.1 While I agree that good animal handling and management skills/knowledge are important for optimal animal welfare, this is not by any means specific to VF and should be made clearer that good animal practices are going to apply to any type of system where livestock are managed.

Line 327: maybe direct quotes should be indicated by italics or quotations to make it clearer the grammatical errors are intentional, and also clearer what are direct quotes from the interviewees.

Lines 361-362: Are you stating that in UK systems, a VF better allow animals to be managed extensively? That the VF technology enables cows to access trees, shade etc, that standard physical fencing doesn’t? Otherwise how is this specific to VF. I know it was mentioned a bit in the introduction, but wanted to double-check as it otherwise doesn’t quite make sense to me as I would think cattle can have trees, shade, scratching opportunities etc without a VF.

Lines 408-413: But there are training protocols for other companies supplying these products. Is there no training available in the UK when implementing Nofence devices?

Line 420: the whole point of the audio cue is that the cattle learn it and respond to it without the need for a visual cue. Is this more about any conflict between cues and social interactions? I don’t think cattle are incapable of learning the audio cue, so trying to better understand what the issue is in this part of the manuscript.

Line 494: is the Kearton et al 2020 ref in the reference list? And what is the Lee and Campbell 2018 reference? Please double-check, I have not seen yet any published evidence that there are actually animals in the studies that do not learn the audio/shock association.

Line 503:  should this be ‘latter’

Line 504: there have been considerable advances in technology since 1981. How relevant is this paper?

Line 518: but if there are visual lines then it defeats the purpose of the GPS technology operating on an audio cue as the association stimulus so that the fences can be easily moved. Then you are back to fixed fencing, or buried wire VFs.

Line 523: but Vence and eShepherd do use physical fences in their training protocols.

Line 560: is the language partly because it is a different sensation when the electrical stimulus is applied directly onto the animal versus from a fence that is grounded?

Line 599: 2019 not 2029

Line 612: But there are also some individuals that don’t respect physical fences and keep pushing through them. VF is not claimed to be 100% containment, nor should it ever be a hard boundary, it is meant for internal fencing with the expectation that some animals may not respect the fence and it would be recommended those animals are removed. Just the same as producers remove animals that continually don’t respect physical electric fences.

Section 3.9

As mentioned before, this is a key point to consider, that we don’t really know how many shocks animals get from physical electric fences. This should not be ignored when so much of this paper centres around the shocks they get from VFs.

Line 703: please check grammar, is ‘valuable’ the correct word?

Line 839: what is this eShepherd ref? As far as I could see this was a company that producers the devices, but this is not who operates eShepherd. They are run out of Gallagher.

English is fine. Writing is clear. A few minor grammatical issues flagged in comments. 

Author Response

Dear Reviewer 1

Thank you very much for your very helpful and detailed feedback.

Attached is a word document which details the changes we have made and answers your points.

If anything is unclear please do not hesitate to let us know.

Kind regards

Iris McCormick

Reviewer 2 Report

The Current paper provides some interesting insight on using GPS shock collars and emphasises the need to better understand the practical implications of them with growing popularity. My suggestions are mainly to improve clarity

Introduction

It would be useful to explain how GPS shock collars work somewhere in here to set up the welfare issues

Could be more concise in areaa

Line 14- “This study provides evidence that…..

Line 18- The use of VF’s in a restricted setting poses a significant risk to livestock welfare.

Line 34-38- This is a long sentence and doesn’t make sense. Please edit

Line 64- Try to avoid starting a sentence with an abbreviation

Line 65-67- How does this improve soil health and sward recovery?

Line 69. Pasture allocation to be modified?

Line 71.- It may help to briefly describe the 5 freedoms

Line 88. Add dash between GPS-SCs

Line 92-96. This is a big sentence. Please refine

Line 151. Please write the abbreviation in full when starting a sentence

Line 183. Delete sentence as it does not add anything

Line 205-210. This may be better after line 200

Table 1. Please provide greater depth in the table heading. Font size changes in the first column. Please define the codes here or move the text in line 158

Discussion

Line 268, 270, 275 and throughout results and discussion- Quotation marks needed. This will help the reader to better identify what the paper is saying from a participant quote

Line 419. Half the participants (n=4)

Line 475 - GPS-SCs

Line 502. The phrasing of this line suggests the study compared the two fence types but it only looked at visual. Please edit

Line 507. Here and in a few places the authors have stated “most participants” however, four participants is only half. This should be edited to half of the participants.

References

Please correct the referencing style to Animals

Please use quotation marks where required

Author Response

Dear Reviewer 2

Thank you very much for your helpful and thoughtful feedback.  We have attached a word document with detailed updates to each point.

Kind regards

Iris McCormick

Reviewer 3 Report

Line 4: please delete the dot at the end of the title

Lines 587 and 739: you cite Lupyan et al. 2020 but there is no position in References. Did you mean the review from Trends in Cognitive Sciences?

I would like to congratulate the Authors a great scientific work and thank for  interesting reading.

Author Response

Dear Reviewer 3

Thank you for your review and support for our manuscript.

The paper we referred to in the language section of the results was

Lupyan, Gary, Rasha Abdel Rahman, Lera Boroditsky, and Andy Clark. "Effects of language on visual perception." Trends in cognitive sciences 24, no. 11 (2020): 930-944.

We have deleted the full stop - thank you!